# Thermal Properties of Illite-Zeolite Mixtures up to 1100 °C

**DOI:** 10.3390/ma15093029

**Published:** 2022-04-21

**Authors:** Štefan Csáki, Ivana Sunitrová, František Lukáč, Grzegorz Łagód, Anton Trník

**Affiliations:** 1Department of Physics, Faculty of Natural Sciences and Informatics, Constantine the Philosopher University in Nitra, Tr. A. Hlinku 1, 94974 Nitra, Slovakia; scsaki@ukf.sk (Š.C.); ivka.sunitrova@gmail.com (I.S.); 2Institute of Plasma Physics of the Czech Academy of Sciences, Za Slovankou 3, 18200 Prague 8, Czech Republic; lukac@ipp.cas.cz; 3Faculty of Mathematics and Physics, Charles University, V Holešovičkách 2, 180000 Prague 8, Czech Republic; 4Faculty of Environmental Engineering, Lublin University of Technology, Nadbystrzycka 40B, 20-618 Lublin, Poland; g.lagod@pollub.pl; 5Department of Materials Engineering and Chemistry, Faculty of Civil Engineering, Czech Technical University in Prague, Thákurova 7, 16629 Prague, Czech Republic

**Keywords:** illitic clay, zeolite, clinoptilolite, thermal expansion, thermal diffusivity, thermal conductivity, specific heat capacity

## Abstract

Illitic clays are the commonly used material in building ceramics. Zeolites are microporous, hydrated crystalline aluminosilicates, they are widely used due to their structure and absorption properties. In this study, illitic clay (Füzérradvány, Hungary) was mixed with natural zeolite (Nižný Hrabovec, Slovakia) with up to 50 wt.% of zeolite content. The samples were submitted to thermal analyses, such as differential thermal analysis, differential scanning calorimetry, thermogravimetry, and dilatometry. In addition, the evolution of thermal diffusivity, thermal conductivity, and specific heat capacity in the heating stage of firing were measured and discussed. The amount of the physically bound water in the samples increased along with the amount of zeolite. The temperature of the illite dehydroxylation (peak temperature) was slightly shifted to lower temperatures, from 609 °C to 575 °C (for sample IZ50). On the other hand, the mass loss and the shrinkage of the samples significantly increased with the zeolite content in the samples. Sample IZ50 reached 10.8% shrinkage, while the sample prepared only from the illitic clay contracted by 5.8%. Nevertheless, the temperature of the beginning of the sintering (taken from the dilatometric curves) decreased from 1021 °C (for illitic clay) to 1005 °C (for IZ50). The thermal diffusivity and thermal conductivity values decreased as the amount of zeolite increased in the samples, thus showing promising thermal insulating properties.

## 1. Introduction

Illite is one of the most abundant clay minerals which is widely used in the production of traditional ceramic materials, including pottery, tiles, etc. [1]. The illitic clay used in this study was mined in northeastern Hungary and its dominant mineral phase is illite. The structure of illite is built up of three sheets—an octahedral sheet sandwiched between two tetrahedral sheets. Between each T-O-T an interlayer cation is placed, which, in most cases, is a potassium cation [2,3,4]. However, some sites are unoccupied, and thus, water molecules populate those vacancies. Depending on the arrangement of the octahedral cations, the illite has several polytypes. The polytype of the studied illite was identified as 1M [5] and its chemical formula can be expressed as:$$K_{0.78}{\mathrm{Ca}}_{0.02}\left({\mathrm{Mg}}_{0.34}{\mathrm{Al}}_{1.69}{\mathrm{Fe}}_{0.02}^{\mathrm{III}}\right)\left[{\mathrm{Si}}_{3.35}{\mathrm{Al}}_{0.65}\right]O_{10}{\left(\mathrm{OH}\right)}_{2}\cdot {\mathrm{nH}}_{2}O$$

Illite undergoes several physicochemical processes during heating, namely evaporation of the physically bound water, dehydroxylation, and sintering. The X-ray diffraction reflections of illite remain observable up to the temperature of 850 °C [6,7].

Zeolites are crystalline, microporous, hydrated, aluminosilicates containing alkali metals or alkaline earth metals. They crystallize in the monoclinic crystal system. The structure is built up from [SiO_4_]^4−^ and [AlO_4_]^5−^ tetrahedrons, which are interconnected by oxygen ions [8,9,10,11]. In the three-dimensional framework, there is a system of channels or cage-like cavities, where alkali cations (K^+^, Na^+^, Ca^2+^) or small molecules, such as H_2_O are placed [12,13,14]. Zeolites have excellent cation exchange properties, thus they are suitable for water and wastewater treatment [9,15,16,17,18]. Moreover, zeolites are widely used in agriculture, building industry, rubber industry, chemical industry, paper industry, households and also in medicine [12,19,20,21,22,23,24]. Zeolites can be divided into several family groups. Mineral clinoptilolite is from the family group of heulandites. This mineral group includes minerals, such as heulandites and clinoptilolite, both of which are crystallized in a monoclinic system and their chemical composition is very similar. The difference between these minerals is in the Si/Al ratio and this affects the thermal stability. The structure of clinoptilolite is destroyed above the temperature of 1000 °C, where amorphization occurs.

Several papers have been devoted to the study of the mechanical, thermophysical, and electric properties of illitic clay and its mixtures with different additions (e.g., fly ash [25,26,27], calcite [28]). Nonetheless, according to the authors’ knowledge, there is no data about the mixtures of illitic clay and natural zeolite. Blends of kaolin and natural zeolite were investigated in [29]. It was shown that zeolite supports the densification of a ceramic body, thus shifting the sintering to lower temperatures.

The aim of this study was to investigate the feasibility of illite-zeolite mixtures and to describe the processes occurring in the mixtures during firing in terms of differential thermal analysis, thermogravimetry, and thermodilatometry. In addition, the thermal diffusivity, conductivity, and specific heat capacity of the prepared samples were measured. Zeolite, being a microporous aluminosilicate is expected to lower the sintering temperature and improve the thermal properties compared with the pure illitic clay. The improvement in the thermal properties is expected through an increase in the porosity of the products due to the addition of zeolite.

## 2. Materials and Methods

Illitic clay was mined near the town of Füzérradvány in northeastern Hungary. The mineralogical composition of illitic clay is as follows: illite (80 wt.%), quartz (12 wt.%), montmorillonite (4 wt.%), and feldspar (4 wt.%) [30]. Zeolite was received from the company ZEOCEM, a.s. (Nižný Hrabovec, Slovakia) as product ZeoBau Micro 50 [31]. The dominant mineral phase in the zeolite sample was Na-clinoptilolite (58.2 wt.%). In addition, cristobalite (12.2 wt.%), illite with mica and albite (9.6 wt.%), quartz (0.7 wt.%), and amorphous phase (19.3 wt.%) were identified as impurities in the sample [22]. The chemical composition of the samples is presented in Table 1.

Zeolite was obtained in a powder form with a diameter of particles <50 μm, while the illitic clay had diverse particle sizes, from several μm up to the cm range. Thus, further processing of the clay was needed in terms of mechanical treatment—starting with the crushing and milling the illitic clay. After crushing, the clay was dried for 2 h at 110 °C. The milling was conducted for 3 min at 350 rpm in the Retsch PM100 planetary ball mill. In the next step, the as-prepared powder was sieved to obtain a powder with particle sizes <100 μm.

The mixtures were prepared by adding 10, 20, 30, 40, and 50 wt.% of zeolite powder to the illitic clay by dry mixing. The studied samples were labeled as IZ10, IZ20, IZ30, IZ40, and IZ50, according to the natural zeolite content, whereas the pure illitic clay and zeolite samples were labeled as ILB and ZEO (see Table 2). To obtain a plastic mass, the green material was mixed with distilled water and the samples were prepared with the help of a laboratory extruder (Ø = 14 mm and *l* = 150 mm). The samples were then dried in the air under laboratory conditions for one week. The samples for thermal analyses, as well as for the microstructure observations were adapted in the final step. Cylindrical samples of dimensions Ø = 14 mm and *l* = 16 mm were prepared for differential thermal analysis and with dimensions of Ø = 14 mm and *l* = 30 mm for thermodilatometry. Disc samples were prepared for thermal diffusivity measurements with a thickness of 2.5 mm and a diameter of 12.5 mm. Differential scanning calorimetry was carried out on powder samples.

Differential thermal analysis (DTA) and thermogravimetry (TG) were performed using the upgraded Derivatograph 1100° (MOM Budapest, Budapest, Hungary) [32]. The measurements were carried out in a static air atmosphere from room temperature up to 1050 °C with a heating rate of 5 °C/min. Bulk alumina was used as the reference material.

Differential scanning calorimetry (DSC) was carried out using a Netzsch Pegasus 404 F3 (NETZSCH Holding, Selb, Germany) in a dynamic Ar atmosphere with a flow rate of 40 mL/min from room temperature to 1100 °C with a heating rate of 5 °C/min.

Thermodilatometry (TDA) was performed on a laboratory made dilatometer from room temperature to 1100 °C with a heating rate of 5 °C/min in a static air atmosphere.

The bulk density was calculated from thermogravimetric and thermodilatometric results to obtain its actual values during firing, according to the following equation:(1)$$\rho =\rho_{0}\frac{\Delta m/m_{0}}{{\left(1+\Delta l/l_{0}\right)}^{3}}\;,$$
where *ρ*_0_ is the bulk density of green samples at room temperature.

The open porosity was calculated with the help of the experimentally determined bulk density and matrix density. The bulk density was obtained from the volume and mass of the cylindrical samples. The matrix density was measured using helium pycnometry (Pycnomatic ATC, Thermo Fisher Scientific, Waltham, MA, USA).

The thermal diffusivity (*a*) measurements were performed on a Linseis LFA 1000 (Linseis Messgeraete GmbH, Selb, Germany) up to 900 °C. Measurements were taken in 100 °C steps, 5 shots per sample. Heat capacity (*c_p_*) was measured simultaneously with the thermal diffusivity. Alumina was used as a reference material. The thermal conductivity (*λ*) was then calculated for each shot using the equation
(2)$$\lambda =\rho \times c_{p}\times a.$$

The measurement was carried out in a vacuum. To achieve equal adsorption and emission properties of all samples, their faces were covered with a thin layer of graphite.

Phase compositions of samples were determined by powder X-ray diffraction (PXRD) methods. The measurements were carried out on vertical powder θ-θ diffractometer D8 Discover (Bruker AXS, Karlsruhe, Germany) using Cu Kα radiation with a Ni Kβ filter. The diffracted beam was detected by the 1D detector LynxEye. Bragg–Brentano geometry was employed with a 0.5 deg fixed divergent slit in the primary beam. The angular range was from 10 to 100° 2θ, step size 0.03° 2θ and the total time in each step was 192 s (the 1D detector is composed of 192 point detectors). Phase identification was conducted using X’Pert HighScore program which accessed the PDF-2 database of crystalline phases. Quantitative Rietveld refinement was performed in TOPAS V5, aiming at the determination of wt.% of all the identified phases following the theory from [33,34]. Small texture correction was included in order to improve the intensities of reflections, see the March–Dollase approach for details [35]. The size of coherently diffracting domains (CDD) and microstrains was evaluated by quantitative Rietveld analysis from the broadening of diffraction peaks using TOPAS V5 software (Bruker AXS, Karlsruhe, Germany). It was assumed that small crystallites and microstrains contribute to the broadening of Lorentzian and Gaussian components of pseudo-Voigt function, respectively [36].

## 3. Results and Discussion

### 3.1. Differential Thermal Analysis

TA of the zeolite sample exhibits a significant endothermic peak in the temperature range from room temperature to ~400 °C (Figure 1a). This peak represented the evaporation of the physically bound water from the pores and the crystal surfaces [21,37]. Similarly, the illitic clay manifests an endothermic peak at temperatures up to ~300 °C, also representing the liberation of the physically bound water. However, this reaction consumes less energy in the illitic clay than in the zeolite sample. The physically bound water is present in the illite in three layers, thus its removal also occurs in three consecutive steps [38]. The process became more emphasized as the part of zeolite increased in the mixture. Moreover, the offset of the dehydration peak gradually shifted to higher temperatures as the zeolite content in the samples increased, in hand with the reaction enthalpy of the process. Once the dehydration process was finished, the dehydroxylation of the illite started (around the temperature of 400 °C).

The dehydroxylation of illite is a two-step process; in the first step, the chemically bound water is removed from the *cis*-vacant illite sites. This is followed by the second step, running at higher temperatures, where the OH^−^ groups are liberated from the *trans*-vacant illite structure [4]. The increasing zeolite amount affected this process as well: as its amount increased, the peaks became less significant, and their positions shifted to lower temperatures (Figure 1b). The first step of the dehydroxylation (peak temperature) was shifted from 608 °C (for illitic clay) to 573 °C (for IZ50). The peak temperature of the second step of dehydroxylation decreased from 714 °C (for illitic clay) to 700 °C (for IZ50). The decreasing peak temperature of the dehydroxylation was not evident in all cases. Sample IZ10 exhibited little higher dehydroxylation temperatures compared to the illitic clay (~by 10 °C in both steps of the process). As the dehydroxylation finished, sintering assisted by viscous flow took place. The shift of the endothermic peak corresponding to the crystallization of new mineral phases was negligible. However, natural zeolite serves as a fluxing agent; thus, the sintering temperatures of the ceramic bodies should be shifted to lower values [33,34].

### 3.2. Differential Scanning Calorimetry

DSC of the prepared mixtures and initial materials (illitic clay and zeolite) is presented in Figure 2. The same features are observable on the DSC curves in the initial stage of heating as on the DTA records. However, due to the higher sensitivity of the DSC apparatus, two-step evaporation of the physically bound water is observable up to 250 °C on the illitic clay curve. In the first step, the weakly bound physically bound water is removed from the crystal surfaces. The second step corresponds to the liberation of tightly bound adsorption water [39]. This feature remained observable, even if the sample contained 50 wt.% of zeolite. With an increase in the temperature, the next step took place (500–700 °C), i.e., the illite dehydroxylation. The two-step character of the process is clearly visible on all curves belonging to the samples containing illite. The temperature shift of the dehydroxylation was not observed on the DSC curves. There were no observed processes in the zeolite sample, except the dehydration at low temperatures and the amorphous phase formation in hand with the sintering process. The structure of the zeolite was destroyed at temperatures above 850 °C (the endothermic peak on the DSC curve) [13,40]. Sintering of the samples containing illite started at higher temperatures, namely around 1000 °C.

### 3.3. Mass Change

The zeolite sample exhibited a continuous decrease in its mass from room temperature up to ~800 °C (Figure 3a). This decrease can be related to the continuous release of the entrapped water molecules [41]. The overall mass loss of the sample reached 11.36% at 1050 °C. The sample prepared from illitic clay exhibited two main mass losses—the first in the temperature interval from room temperature up to ~300 °C, where the physically bound water was evaporated from the sample. The mass loss reached 2.8% in this region. The mixtures containing the zeolite exhibited higher mass losses. Thus, the mass losses of the mixtures gradually increased from 3.5% (for IZ10) to 5.3% (for IZ50). The next significant mass loss was observed during the illite dehydroxylation. For the illitic clay sample, the decrease in the sample mass reached 3.9% over the temperature interval from 450 °C to 750 °C. The mass loss in the discussed region gradually decreased, as the zeolite content in the sample increased. This feature was caused by the fact that zeolite does not undergo dehydroxylation, as it does not contain OH^−^ groups in the structure. The overall mass losses increased along with zeolite content. For the illitic clay sample, the value of the mass loss reached 6.9%. A comparison of the mass losses of the individual samples is visualized in Figure 3b. The addition of zeolite significantly increased the overall mass loss. The mass loss of the sample IZ50 increased by 9.5%, which was almost 3% higher than that of the illitic clay.

### 3.4. Thermal Expansion

The zeolite sample suffered a slight contraction up to ~870 °C (Figure 4a). This contraction reached 1.9% and can be related to the removal of the water molecules from the zeolite cavities and capillaries. At ~870 °C, a progressive shrinkage of the zeolite sample started, which indicated the start of the sintering and appearance of the glassy phase. Moreover, the zeolite structure collapsed at the above-mentioned temperature and an amorphous phase was formed. The overall shrinkage of the sample reached 14.5% at 1100 °C.

The sample prepared from the illitic clay exhibited a continuous thermal expansion up to the start of the dehydroxylation process (~470 °C). This expansion reached 0.3%. During the dehydroxylation, and due to the *α*→*β* transformation of quartz, the expansion became more pronounced, reaching 1.3% at 690 °C. This expansion is thus the superposition of the two competitive processes. The *α*→*β* transformation of quartz brings about around 0.68% expansion [42]. During dehydroxylation, the ring of the tetrahedral sheet of the illite expands to allow the water molecules to pass, which contributed to the expansion as well. The two-step character of the illite dehydroxylation was also observable on the temperature dependence of the coefficient of linear thermal expansion curves (Figure 5), where two distinct peaks marked the process. Once the dehydroxylation was finished, a continuous expansion of the illitic clay sample was observed. This is related to the expansion of the *b* and *c* axes and the simultaneous shrinkage of the *a* axis and *β* angle of the dehydroxylated illite crystals [7]. At ~920 °C, a progressive shrinkage started marking the start of the sintering assisted by viscous flow. The overall shrinkage reached 5.6%.

The TDA curves of the mixtures exhibit similar features to those of the illitic clay. The influence of the zeolite addition, regardless of its amount, was not significant up to the start of the sintering; the expansion of the samples did not differ at temperatures up to 400 °C (0.27%). Before the start of the sintering process, the relative length changes of the samples reached 1.54% and 1.31% for the illitic clay and the IZ50 sample, respectively. Thus, the dimension changes of the samples were mainly driven by the illite and the contraction of the zeolite was suppressed. The start of the sintering (taken as the onset of the TDA curves) was shifted to lower temperatures with increasing zeolite content (inset figure in Figure 4) from 1020 °C (for illitic clay) to 1005 °C (for IZ50). This can be related to the good fluxing properties of the natural zeolite [33,34]. The overall shrinkage of the samples increased from 5.6% (for illitic clay) to 10.8% (for IZ50), as shown in Figure 4b.

### 3.5. Phase Composition

The phase composition of the fired illite-zeolite samples (Figure 6 and Table 3) manifested significant differences. Quartz and Al_2_O_3_ were identified as dominant mineral phases in the fired illitic clay sample supported by orthoclase, mullite and amorphous phase. On the other hand, the fired zeolite sample contained cristobalite and albite as dominant phases supported by quartz and amorphous phase. The amorphous content in both samples is comparable.

XRD analysis (Figure 6 and Table 3) of the samples revealed a decreasing quartz content with an increasing amount of zeolite in the samples. On the other hand, the amount of cristobalite gradually increased from 3 wt.% (sample IZ10) to 11 wt.% (sample IZ50). Orthoclase (was forming due to the high potassium content in the illitic clay), KAlSi_3_O_8_, disappeared when the amount of zeolite in the samples exceeded 30 wt.%, being replaced by albite (NaAlSi_3_O_8_). As zeolite contains a higher amount of Na, if this exceeds a certain value, albite is formed instead of orthoclase. The amorphous content of all the samples is similar, almost within the experimental error.

### 3.6. Bulk Density

Temperature dependence of the bulk density (Figure 7) of the illitic sample was influenced by the physicochemical processes running in the illite. The removal of the physically bound water led to a smooth step-like decrease in the bulk density up to the temperature of 200 °C. After this process was finished, a continuous decrease in the bulk density was observed until the dehydroxylation started. As water molecules possess a higher density than clay, their removal leads to a decrease in the bulk density. After the dehydroxylation, a minimum of the bulk density was reached, around the temperature of 900 °C. As the sintering started, significant densification could be observed. The bulk density of the sample increased by ~10%. Compared with the green illitic sample density (~1800 kg/m^3^) the density of the pure zeolite sample was considerably lower (~1200 kg/m^3^). The dehydration of the zeolite led to a continuous decrease in the density of the sample up to ~400 °C, where a plateau was observed. The bulk density of the zeolite sample exhibited a significant increase as the sintering started above 850 °C. The sintering led to a ~30% increase in the bulk density of the zeolite sample.

Temperature dependence of the bulk density of the mixtures exhibited similar features as those of the illitic clay. However, differences were found in the degree of densification. Zeolite supported the densification; thus, with an increasing amount of zeolite in the samples, the relative increase in the bulk density between room temperature and the maximum sintering temperature (1100 °C) increased. Sample IZ10 exhibited only a 10% increase in the value of bulk density. However, this increased to 14% for IZ20, 16% for IZ30, 19% for IZ40, and 22% for IZ50. A similar trend was observed in [29] for the mixtures of kaolin with zeolite. Nevertheless, the final values of the bulk density remained lower, as kaolin exhibited lower bulk density and the densification of the illitic clay was more progressive than that of kaolin.

### 3.7. Porosity

Initial values of the porosities (Figure 8) of the green samples were stretched from 36% (illitic clay and IZ10) to 49% (zeolite). Upon heating, the value of the porosity of the samples exhibited a gradual increase due to the removal of the physically and structurally bound water. As the water molecules were escaping, the sample voids were left behind. The increase stopped as the sintering of the samples started, which was observed to take place from 900 °C. The progressive sintering of the samples was evident—the values of the porosities of all samples decreased below 8%, except the zeolite, which exhibited the porosity values of 17%. The lowest value of porosity was observed in the case of the illitic clay sample, for which the final value reached 2.2%. However, sample IZ10 also exhibited excellent densification behavior, as its porosity remained similar to that of the illitic clay sample. All the other samples reached higher values than those of the illitic clay or IZ10.

### 3.8. Thermal Diffusivity

Thermal diffusivity of the mixtures exhibited the highest values at room temperature (Figure 9a) due to the increased water content of the samples (the physically bound water has not been removed yet at this temperature). The highest value was found for sample IZ10 and with increasing zeolite content, a gradual increase in the diffusivity value was observed (Figure 6, right)—thermal diffusivity of sample IZ50 reached only 0.35 mm^2^/s, while the IZ10 sample had a thermal diffusivity of 0.54 mm^2^/s. With increasing temperature, the physically bound water was gradually removed, and the values of thermal diffusivity decreased. The decrease became less significant after the physically bound water was liberated. During dehydroxylation, the thermal diffusivity reached its minimum due to the creation of a highly defective structure of illite (octahedral layers losing their OH^−^ groups). Once the dehydroxylation finished, the thermal diffusivity started to increase along with the increasing temperature. The measurements were carried out only up to 900 °C due to the progressive shrinkage of the samples during sintering (the samples could not remain in the sample holder). The highest decrease and the most pronounced dependence of the diffusivity values on the temperature was observed in the case of the IZ10 sample (the value of the diffusivity decreased by 21%).

Zeolite exhibits almost constant values of thermal diffusivity over the whole studied temperature interval (0.21 mm^2^/s) [29]. With increasing zeolite content, the dependence of the thermal diffusivity on the temperature became less significant. The lowest values of thermal diffusivity were observed for the IZ50 sample almost in the whole studied temperature interval. Moreover, the thermal diffusivity of sample IZ50 changed only slightly during the heating. The differences in the thermal diffusivity values of the individual samples decreased with increasing temperature (Figure 9b).

### 3.9. Thermal Conductivity

The thermal conductivity values follow the same trend as the thermal diffusivity values—with increasing zeolite content, the value of thermal conductivity decreases (Figure 10a). The effect of the physically bound water was pronounced in sample IZ10, whilst the other samples exhibited almost no changes in the thermal conductivity values up to 300–400 °C. In this temperature region, the thermal conductivities of all samples reached their minimal values. The plateau-like dependency of the thermal conductivities of the samples can be ascribed to the gradual evaporation of water from the zeolite in the samples, which prevents the decrease in the thermal conductivity after the evaporation of the physically bound water from the illite. As the dehydroxylation started the thermal conductivity began to increase and this lasted up to 900 °C. It should be noted that measurements were finished at the onset of the sintering to prevent the samples from falling from the holder due to their progressive shrinkage. Comparing the thermal conductivity values of individual samples showed that the zeolite addition led to a decrease at all temperatures (Figure 10b). This behavior can be explained by the increased porosity of the samples with higher zeolite content. In addition, the thermal conductivity of zeolite is only 0.37 W/(m·K) [29]. Thus, its increasing amount decreases the thermal conductivity. At room temperature, the IZ10 sample exhibited a conductivity value of 1.03 W/(m·K), while the sample containing 50 wt.% of zeolite reached only 0.50 W/(m·K), half of that of the IZ10 sample. At 900 °C, this difference decreased—the values of thermal conductivity were 1.90 W/(m·K) and 1.47 W/(m·K) for the IZ10 and IZ50 samples, respectively. Nevertheless, the difference still remained significant. On the other hand, it should be mentioned, that the uncertainty of the measurement method reaches 6%. This feature suggested that the mixture has potential in the building industry, as the thermal insulation properties are improving with an increasing zeolite addition.

### 3.10. Specific Heat Capacity

Heat capacity (Figure 11) exhibited an increasing trend with rising temperature, as expected by the Debye model. The addition of zeolite did not lead to any significant difference in the specific heat capacity at room temperature. The values of the specific heat capacity fall within the range from 1.12 to 1.02 kJ/(kg·K), in decreasing order with increasing zeolite content. At higher temperatures, the trends were not that straightforward. The effect of the removal of the physically bound water, as well as the effect of dehydroxylation, were not observable on the temperature dependence of the specific heat capacity. Nevertheless, the differences became significant, as the temperature reached 800 °C, and remained similar at 900 °C, where the sintering started. An opposite behavior was observed in the case of the mixtures of kaolin with zeolite, where the specific heat capacity decreased with an increasing zeolite content [29]. At the temperature of 900 °C, the value of specific heat capacity reached 2.97 kJ/(kg·K) for the IZ10 sample and 3.76 kJ/(kg·K) for the IZ50 sample.

## 4. Conclusions

The mixtures of illitic clay (originating from northeastern Hungary) with five different amounts of natural zeolite (Slovakia) were studied by thermal analyses. Differential thermal analysis, thermogravimetry, and differential compensating calorimetry showed that the addition of zeolite influences the amount of physically bound water in the samples. Moreover, the addition of zeolite decreased the peak temperatures of the illite dehydroxylation—from 609 °C to 575 °C in the first step and from 713 °C to 702 °C in the second step. The overall mass losses of the mixtures changed significantly with an increasing zeolite amount. The illitic clay exhibited 6.9% mass loss over the studied temperature region, while the mass loss of the IZ50 sample reached 9.5%. The zeolite addition decreased the onset temperature of the sintering process from 1021 °C (illitic clay) to 1005 °C (IZ50) and increased the sample shrinkage from 5.6% (illitic clay) to 10.8% (IZ50). The dominant mineralogical phase in the IZ10 sample was quartz. However, its amount gradually decreased by increasing the zeolite content. As the amount of zeolite exceeded 30 wt.%, albite started to form instead of orthoclase. This was explained by the sodium content of the zeolite, which supports the formation of albite. Thermal diffusivity and conductivity reached the highest values for the samples with the lowest amount of zeolite (0.54 mm^2^/s and 1.03 W/(m·K) for diffusivity and conductivity, respectively). The difference between the values of the diffusivity of the samples decreased with increasing temperature. The lowest changes in the thermal diffusivity were observed in the IZ50 sample. A minimum in the diffusivity values was achieved around the illite dehydroxylation temperature. The above-mentioned results confirmed that by adding even a low amount of zeolite to the illitic clay, its sintering starts at lower temperatures. Moreover, zeolite addition increased the sample porosity which, in turn, led to a decrease in the values of the thermal conductivity. Due to the lower thermal conductivity values of the samples, the prepared mixtures might find uses as building materials, with better insulating properties, than pure illitic clay.

## Figures and Tables

**Figure 1 materials-15-03029-f001:**
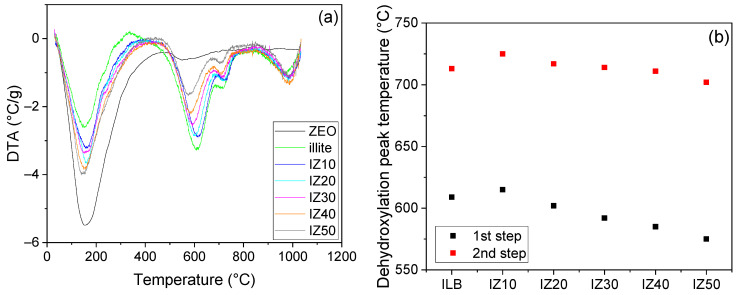
Differential thermal analysis of the samples (**a**). Peak temperatures of the illite dehydroxylation (1st and 2nd step) (**b**).

**Figure 2 materials-15-03029-f002:**
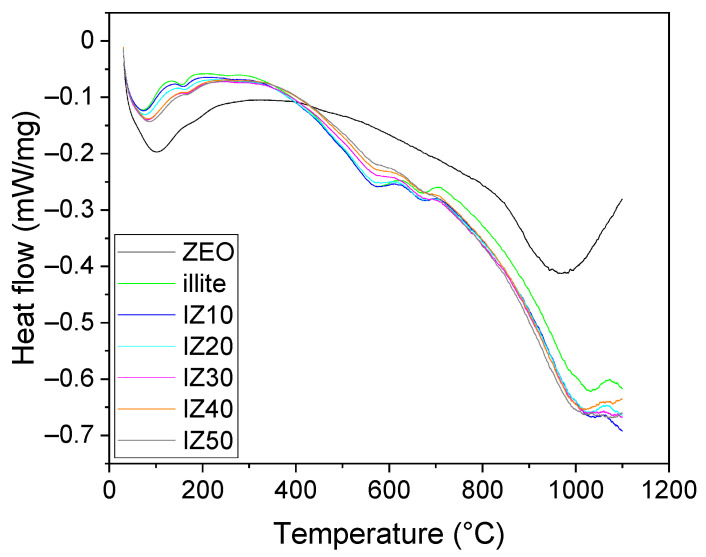
DSC analysis of the mixtures and initial materials.

**Figure 3 materials-15-03029-f003:**
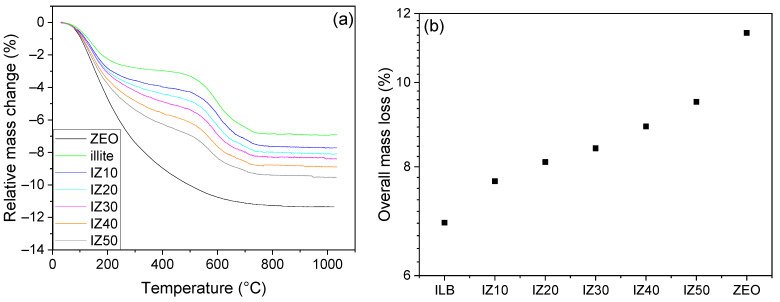
The mass change of the prepared samples and initial materials (**a**). Comparison of the total mass losses (**b**).

**Figure 4 materials-15-03029-f004:**
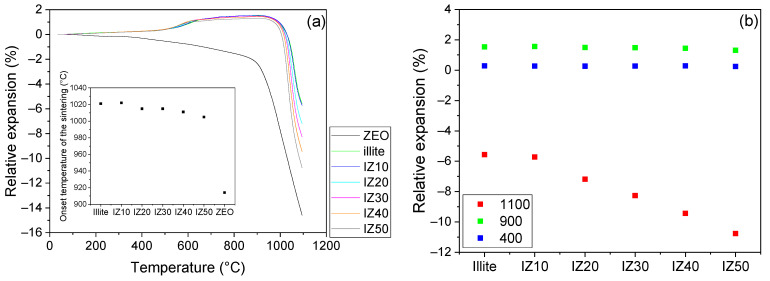
Dilatometry of the prepared mixtures and initial materials. Inset graph: Onset of the sintering (**a**). Relative expansion of the samples at different temperatures (**b**).

**Figure 5 materials-15-03029-f005:**
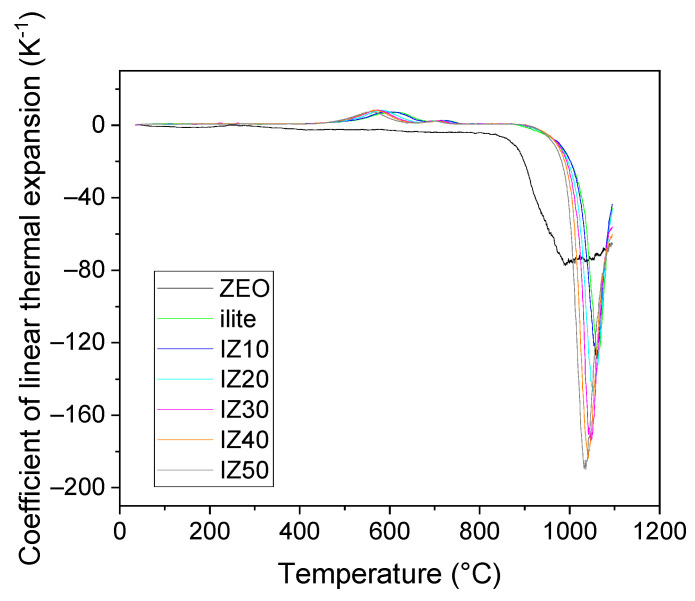
Coefficient of linear thermal expansion of the samples.

**Figure 6 materials-15-03029-f006:**
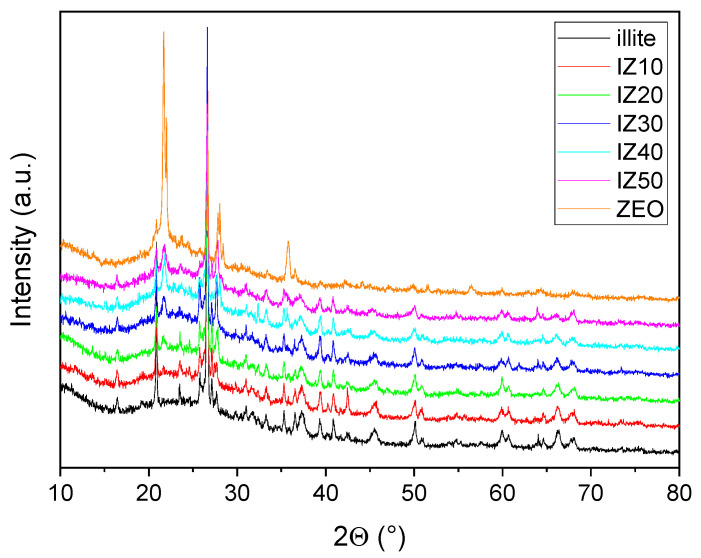
XRD analysis of the prepared samples.

**Figure 7 materials-15-03029-f007:**
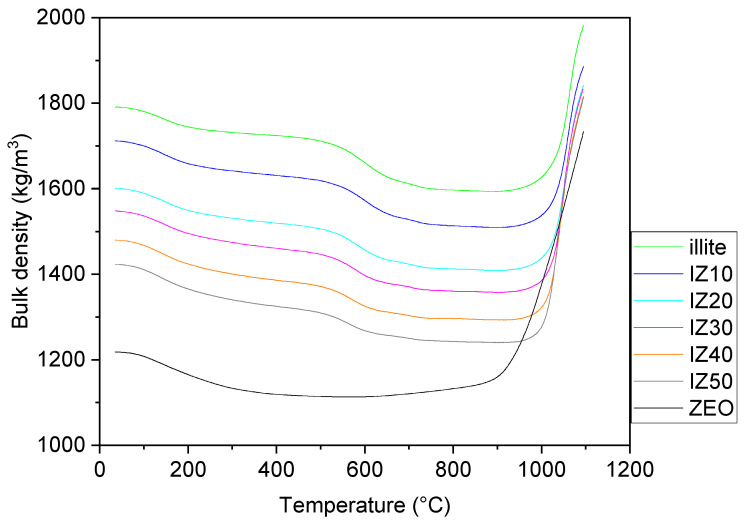
Bulk density of the samples.

**Figure 8 materials-15-03029-f008:**
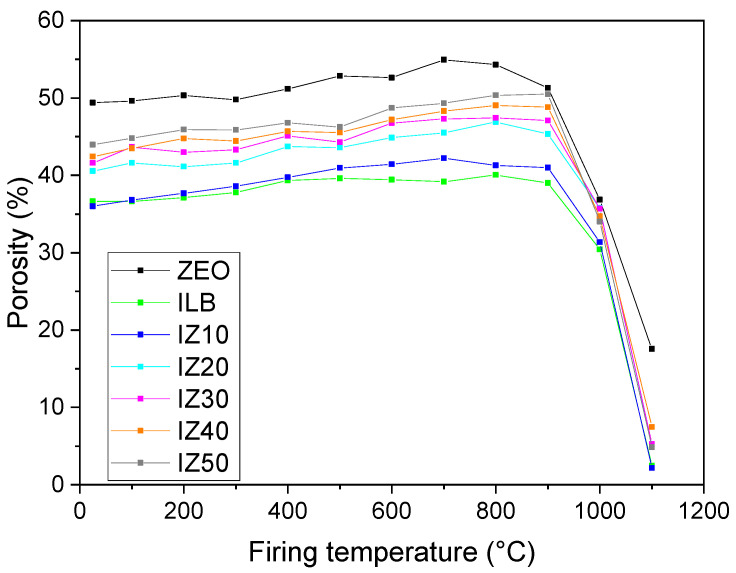
Porosity of the samples fired at different temperatures.

**Figure 9 materials-15-03029-f009:**
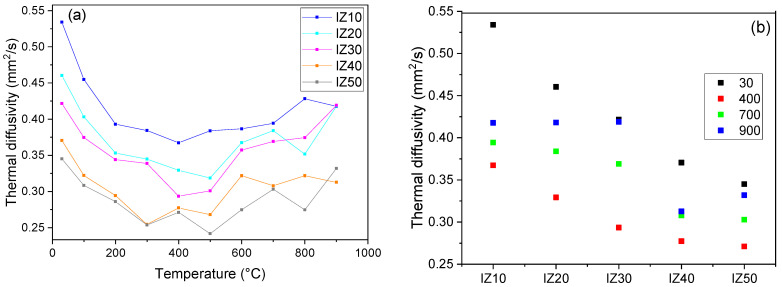
Thermal diffusivity of the mixtures (**a**). Comparison of thermal diffusivities at different temperatures (**b**).

**Figure 10 materials-15-03029-f010:**
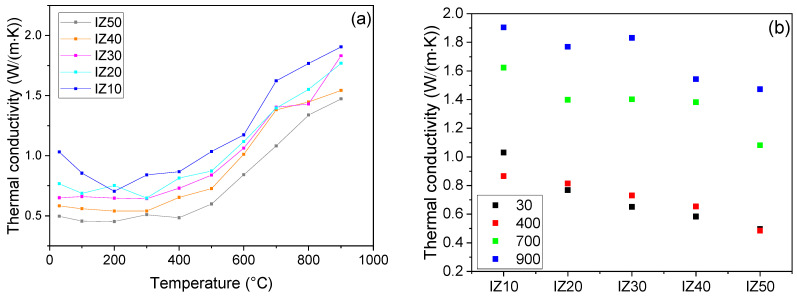
Thermal conductivity of the mixtures (**a**). Comparison of thermal conductivities at different temperatures (**b**).

**Figure 11 materials-15-03029-f011:**
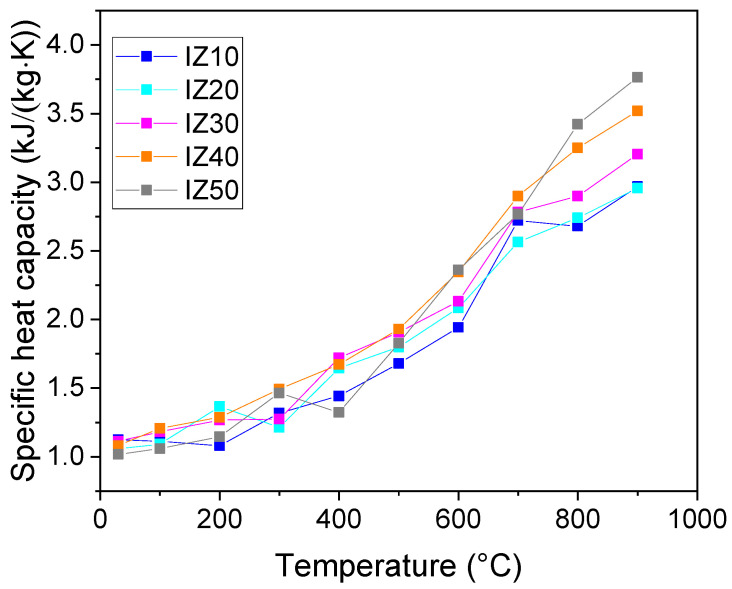
Specific heat capacity of the samples.

**Table 1 materials-15-03029-t001:** Chemical composition of illitic clay and natural zeolite (in wt.%).

Oxides	SiO_2_	Al_2_O_3_	Fe_2_O_3_	TiO_2_	CaO	MgO	K_2_O	Na_2_O	L.O.I
Illitic clay	58.4	23.9	0.6	-	0.4	1.7	7.7	0.1	7.2
Zeolite	68.2	12.3	1.3	0.2	3.9	0.9	2.8	0.7	11.35

**Table 2 materials-15-03029-t002:** The compositions of the samples made from Sedlec kaolin and natural zeolite (in mass%).

Sample	ILB	IZ10	IZ20	IZ30	IZ40	IZ50	ZEO
Illitic clay	100	90	80	70	60	50	-
Zeolite	-	10	20	30	40	50	100

**Table 3 materials-15-03029-t003:** Phase composition of the fired samples (in wt.%).

Mineral Phase	Illitic Clay	IZ10	IZ20	IZ30	IZ40	IZ50	ZEO
Quartz	34	32	28	25	24	22	14
Al_2_O_3_	31	26	27	25	20	21	-
Orthoclase	13	15	17	16	-	-	-
Mullite	22	24	21	26	25	21	-
Cristobalite	-	3	7	8	10	11	52
Albite	-	-	-	-	21	25	34
Amorphous	56	53	47	47	43	47	45

## Data Availability

Not applicable.

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
