# Peer review of "Thermal Properties of Illite-Zeolite Mixtures up to 1100 °C"

_materials, 2022, doi:10.3390/ma15093029_

Round 1

Reviewer 1 Report

Csáki et.al. have provided a study focused on understanding the properties like differential thermal analysis, thermogravimetry, thermodilatometry, thermal diffusivity, conductivity, and specific heat capacity of the Illite-Zeolite Mixtures samples. The reported work may be considered for publication.

The author needs to address the following question:

  • The importance of this work and its impact needs to be described in the introduction.
  • Line 221: It should be “mass loss of the sample IZ50 increased by 9.5%, which was almost 3% higher than that of the illitic clay.”
  • Line 434: “the zeolite addition led to a decrease at all temperatures”. Is there any alternative explanation for this phenomenon?

Author Response

Line 221: It should be “mass loss of the sample IZ50 increased by 9.5%, which was almost 3% higher than that of the illitic clay.”

Answer:

We corrected it.

Line 434: “the zeolite addition led to a decrease at all temperatures”. Is there any alternative explanation for this phenomenon?

Answer:

We added some explanation.

Reviewer 2 Report

The manuscript from Trník and coworkers describes the thermal properties of physical mixtures of Illite and a natural zeolite. The manuscript follows closely the structure of a previous publication of the authors dealing with Kaolin-zeolite blends (ref 29). Although no particular high impact result was obtained, the set of results could be of interest of the readers of this field. Nevertheless, the authors are suggested to consider the following points:

- It is not clear from the introduction, the importance of Illite and how the physical mixture with zeolites is expected to improve its properties. Moreover, along the manuscript most of the results seem to be average values of both materials (considering the relative proportions). If it not the case, it would increase the impact of the work if more emphasis is put in the synergistic effect of the mixture.

-It seems that the thermal conductivity benefits from the mixture with zeolites. At least this point should deserve a deeper discussion. Are the changes significant when compared with other commercial materials? What application do the authors envisage for these materials? In what extent are these materials advantageous?

- Some data interpretation would benefit from the indication of the error associated to the measurements.

- The authors should also be aware that, as indicated, the percentage of zeolite in the natural zeolite samples is far from 100 % and, in fact, these samples have a non-negligible amount of illite. Therefore, the term “zeolite” in the designation of the mixtures should be used with care, because it could mislead the reader about the effective zeolite content in each mixture.

Author Response

- It is not clear from the introduction, the importance of Illite and how the physical mixture with zeolites is expected to improve its properties. Moreover, along the manuscript most of the results seem to be average values of both materials (considering the relative proportions). If it not the case, it would increase the impact of the work if more emphasis is put in the synergistic effect of the mixture.

Answer:

We added some explanation in Introduction.

-It seems that the thermal conductivity benefits from the mixture with zeolites. At least this point should deserve a deeper discussion. Are the changes significant when compared with other commercial materials? What application do the authors envisage for these materials? In what extent are these materials advantageous?

Answer:

We added several explanations in Thermal conductivity.

- Some data interpretation would benefit from the indication of the error associated to the measurements.

Answer:

We added the uncertainty of the measurement in Thermal conductivity.

- The authors should also be aware that, as indicated, the percentage of zeolite in the natural zeolite samples is far from 100 % and, in fact, these samples have a non-negligible amount of illite. Therefore, the term “zeolite” in the designation of the mixtures should be used with care, because it could mislead the reader about the effective zeolite content in each mixture.

Answer:

You are right. Our illite is not pure illite and the same is valid also for natural zeolite. The mineral compositions of illite and zeolite are described in section of Materials and Methods. We use illitic clay instead of illite in the manuscript. For zeolite we use natural zeolite.